# Susceptibility Weighted Imaging in Migraine with and Without Aura: A Case–Control Study

**DOI:** 10.3390/neurolint17070104

**Published:** 2025-07-08

**Authors:** Adrian Scutelnic, Tomas Klail, Diego Moor, Nedelina Slavova, Valentina Petroulia, Simon Jung, Mattia Branca, Urs Fischer, Franz Riederer, Roland Wiest, Christoph J. Schankin

**Affiliations:** 1Department of Neurology, Inselspital, University Hospital Bern, University of Bern, 3010 Bern, Switzerland; adrian.scutelnic@insel.ch (A.S.); diego.moor@students.unibe.ch (D.M.); nedelina.slavova@insel.ch (N.S.); simon.jung@insel.ch (S.J.); urs.fischer@insel.ch (U.F.); franz.riederer@insel.ch (F.R.); 2Institute of Diagnostic and Interventional Neuroradiology, Bern University Hospital, 3010 Bern, Switzerland; tomklail@gmail.com (T.K.); valentina.petroulia@gmail.com (V.P.); roland.wiest@insel.ch (R.W.); 3CTU Bern, University of Bern, 3012 Bern, Switzerland; mattia.branca@unibe.ch; 4Centre for Migraine and Headache, Bellevue Medical Group, 8001 Zürich, Switzerland

**Keywords:** migraine, aura, cortical spreading depression, headache

## Abstract

Background: The asymmetry of cortical veins in susceptibility weighted imaging (SWI) in MRI might be a biomarker for migraine aura and cortical spreading depression (CSD). The aim of this study was to assess in humans if SWI asymmetry can be found in patients who have migraine attacks without aura. Methods: We included patients (*n* = 100 per group) from the emergency room setting when they (i) presented with an acute neurological deficit or headache; (ii) had a discharge diagnosis of a migraine aura, a migraine without an aura, or neither (controls without stroke or epilepsy); and (iii) had a brain MRI with SWI in the acute setting. Results: In the migraine with aura group, SWI asymmetry was found in 26% (95% CI 18–35) compared to patients with migraine without aura (3%, [95% CI 1–8], *p* < 0.001) and controls 7% [95% CI 3–14], *p* < 0.001). There was no difference between patients with migraine without aura and controls (*p* = 0.19). Conclusions: The distinct SWI changes in migraine with and without aura suggest that CSD might not be involved in the pathophysiology of migraine without aura.

## 1. Introduction

There is still uncertainty about whether cortical spreading depression (CSD) causes migraine headache, particularly in patients with migraine without aura [1,2]. While several experimental studies demonstrated the activation of peripheral and/or central nociceptive trigeminal pathways by CSD [3,4,5,6,7,8,9], others have failed to confirm this association [10,11]. The hypothesis that CSD causes migraine headache is challenged by clinical observations, such as different responses to prophylactic treatments of migraine headache compared to aura, the lack of a locked temporal relationship between migraine aura and migraine headache, and the occurrence of migraine aura without headache [1,12,13,14,15].

Susceptibility weighted imaging (SWI) in magnetic resonance imaging (MRI) depicts cerebral veins according to their oxyhemoglobin concentration [16]. In 1944, Leão demonstrated the dilation of pial arteries and veins in rabbits, as well as an increase in oxyhemoglobin in pial veins following a wave of CSD [17]. Asymmetric abnormalities on SWI, i.e., dilated veins, can be found during or shortly after migraine aura and possibly reflect recent CSD [18].

The aim of this study was to assess in humans if SWI asymmetry can be identified in patients who have migraine attacks without aura to test the hypothesis that migraine without aura might also be caused by CSD, which would then be considered ‘silent aura’. For this, we searched for traces of CSD using SWI in emergency room patients who had spontaneous acute migraine attacks without aura. 

## 2. Methods

This is a retrospective case–control study that was approved by the local ethics committee (Bern cantonal ethics committee, 2020-00115, approval date 3 February 2020). The preliminary results were presented as a poster at the European Academy of Neurology in 2023. Patients were recruited as part of a previous study [19]. Included patients had to provide written informed consent, as required by the local regulations. The written general consent allows patient-related data from the routine clinical care to be used in anonymized form for research purposes. In brief, patients with migraine with aura and controls without migraine and aura (*n* = 100 per group, consecutive for each group) were included when they (i) presented at our emergency department with an acute neurological deficit, (ii) had a brain MRI either when symptoms were still present or within eight hours after cessation of symptoms, and (iii) had a discharge diagnosis of migraine with aura or diagnoses other than migraine, acute ischemic stroke, and epileptic seizures (controls). Patients with alternative explanations for SWI changes (e.g., brain tumor and vascular malformation) were excluded. The cut-off of eight hours has been chosen based on previous findings demonstrating persistent dilated cerebral veins within this timeframe in patients with migraine aura [18]. Patients with migraine without aura (*n* = 100) were included when (i) they had a brain MRI performed during the migraine headache and (ii) had a discharge diagnosis of migraine without aura. Due to the exploratory nature of this analysis and lack of data on frequency of SWI changes in migraine without aura, no prior power calculation was performed. 

Clinical information was taken from the discharge letters. For this study, health-related data assessed within the scope of clinical routine examinations and decision making were used. Diagnoses have been made by experienced neurologists in the emergency department who have been trained to identify migraine with and without aura according to the International Classification of Headache Disorders [20]. No imputation for missing data has been performed. AS and CJS had access to all clinical and radiological data. 

The SWI changes were assessed by two neuroradiologists (TK and VP) blinded to the diagnosis. An SWI change was defined as an asymmetry right versus left involving at least two veins (Figure 1). The MRI scanners used had a field strength of 1.5 or 3 Tesla. The SWI slice thickness varied from 1.2 mm to 2.0 mm. SWI images were first visually screened for unilateral prominence of cortical veins by the rating of neuroradiologists. When ≤2 veins appeared prominent, as rated by the expert neuroradiologist, we applied the approach described in the study by Slavova et al. [21]. The maximal diameter of the most prominent vein was measured on the source SWI image and divided by the diameter of its contralateral homolog as originally applied for the definition of index vein. A ratio ≥ 2.0 was considered significant. When >2 veins appeared prominent after visual analysis, the venous asymmetry was graded qualitatively (mild/moderate/marked) by the two blinded neuroradiologists (VP and TK) in the predefined regions as described in previous studies [18,19]. Discrepancies were resolved by consensus.

For the statistical analysis, STATA (StataCorp LCC 1985–2019, College Station, TX, USA) has been used. The categorical data is presented in counts and confidence intervals according to Wilson’s method. Continuous variables are presents as means with standard deviation or medians with interquartile ranges, as appropriate. For comparisons between categorical variables, Chi-square or Fischer’s exact test was used, as appropriate. For comparisons between non-normally distributed continuous variables, Mann–Whitney or Kruskal–Wallis test was performed, as appropriate. For multiple comparisons, Bonferroni correction has been performed to correct for Type I error.

Upon reasonable request, study data will be made available. All methods used in this study were carried out in accordance with the local regulations.

## 3. Results

There were more women in the migraine with and without aura group compared to controls (68 females with migraine with aura, 86 in migraine without aura, and 55 in controls; *P*_Bonferroni_ < 0.001), and patients with migraines with and without aura were younger than controls (39 ± 15 years old in migraine with aura, 36 ± 11 in migraine without aura, and 61 ± 13 in controls; statistics, *P*_Kruskal-Wallis_ < 0.001). In the migraine with aura, migraine without aura, and controls groups, the MRI was performed after a median of 4 (IQR 2.7–5.9), 72 (IQR 26–216) and 4.1 h (IQR 2.9–6.8) after the onset of symptoms. For co-morbidities see Table 1. 

Cohen’s kappa for the SWI assessment was 0.85. In the migraine with aura group, SWI asymmetry was found in 26% (95%CI 18–35), significantly more often than in patients with migraine without aura (3%, 95%CI 1–8, *p* < 0.001) and controls (7%, 95%CI 3–14, *p* < 0.001). There was no difference between patients with migraine without aura and controls (*p* = 0.19). The clinical information of the patients with migraine without aura and controls who had asymmetric SWI is shown in Table 2. In nine patients with a migraine without an aura, MRI was performed within eight hours after the symptom onset. None of these had SWI asymmetry.

In patients with a migraine with aura, only a speech disturbance was associated with the SWI asymmetry (Table 3). 

To assess the impact of the time after the symptoms had ceased on the SWI asymmetry, the frequency of the SWI asymmetry was separately assessed in patients with ongoing and ceased focal neurological symptoms (i.e., in migraine with aura and controls only). In patients with migraine with aura and ongoing symptoms, 29% (8/35) had SWI venous asymmetry compared to 25% (16/65) whose symptoms had ceased prior to the MRI (*p* = 0.84). In controls, the proportions were 6% (5/77) and 8% (2/23), respectively (*p* = 0.65). There was no association between the time from the symptom onset to the MRI and SWI changes (*P*_whole population_ = 0.41, *P*_migraine aura_ = 0.24, *P*_migraine without aura_ = 0.47, *P*_controls_ = 0.76). 

## 4. Discussion

The main finding of our study is the higher frequency of the SWI asymmetry in patients with migraine with aura compared to migraine without aura. Assuming that the SWI changes reflect the CSD, our results suggest that the CSD is not involved in the pathophysiology of migraine without aura [10,11,12,13,14,15,22].

This large-scale study in patients with spontaneous attacks complements a smaller case series using perfusion studies that were not able to show any perfusion changes, i.e., another potential imaging marker suggestive of migraine aura, in migraine without aura in humans [22,23]. In contrast, previous reports demonstrated *symmetric* occipital or global perfusion changes in the headache phase of migraine without aura [24,25,26], which cannot be compared directly with our findings of *asymmetric* SWI alterations. Typically, CSD is a unilateral event [27] and should not result in global or bilateral perfusion changes. In our opinion, therefore, both, the presence of symmetric perfusion abnormalities and the lack of asymmetric SWI alterations are consistent with the absence of CSD in migraine without an aura. 

We found more SWI asymmetry in controls (7%) than in patients with migraine without an aura (3%), although this was not significant (*p* = 0.19). Four of the seven controls with SWI changes had transient global amnesia (TGA). Given the association between TGA and migraine with aura [28], the SWI changes in patients with TGA might have been mediated by mechanisms similar to those involved in migraine aura. Similarly, two of the seven control patients with SWI asymmetry had benign positional vertigo, a condition also associated with the diagnosis of migraine, although the relevance of aura was not reported [29].

In contrast to ‘silent aura’ causing attacks of migraine without aura, diencephalic and brainstem structures might be more important as concluded from functional neuroimaging studies about the premonitory phase of triggered and spontaneous migraine attacks [30,31,32]. Our data aligns with a recent case study indicating that the occurrence of headache in both migraine with and without aura is probably not connected to CSD [32].

One limitation of this study is the long time-lapse between the headache onset and MRI (median 72 h) in migraine without aura compared to the time-lapse between the aura onset and MRI (median 4 h) in the migraine with aura group. There might have been a selection bias of patients with headache of longer durations, given the clinical suspicion of a secondary headache which requires an MRI examination. However, several points argue against this confounding our results: (i) None of the migraine without aura patients who had the MRI within 8 h after headache onset had SWI asymmetry. This would be extremely uncommon if CSD occurred in patients with migraine without aura, since SWI-alterations persist after the CSD for at least 8 h [18]. (ii) Furthermore, the lack of an association between the time of the symptom onset and SWI changes suggests that the SWI changes are longer lasting, making them a reliable finding in assessing migraine aura. (iii) Furthermore, the frequency of SWI changes in migraine with aura and controls increased only marginally and non-significantly after comparing patients with ongoing symptoms to those with symptoms that had ceased during the MRI. This finding, however, needs confirmation—ideally through longitudinal studies involving repeat MRIs at different time points—since previous reports found an association between venous changes assessed by MRI and the time from the symptom onset in migraine with aura [23]. Due to the heterogeneity of daily and migraine abortive medication and a lack of precise documentation of the time point of the medication intake (e.g., of triptans), we did not assess the possible influence of treatments on our findings. Another limitation of our study is its retrospective design and lack of a systematic assessment of clinical data (e.g., of baseline characteristics such as obesity). Due to the inclusion of patients treated in the emergency department, MRI scanners with different field strengths were utilized. Additionally, there was a variation in the slice thickness of the SWI. The main strength is the large number of subjects investigated.

In conclusion, asymmetric SWI changes in migraine without aura occur significantly less frequently than in migraine with aura. Assuming a canonical succession of the aura to the headache, the lack of SWI changes in patients with migraine without aura argues against ‘silent’ auras causing migraine without aura. 

## Figures and Tables

**Figure 1 neurolint-17-00104-f001:**
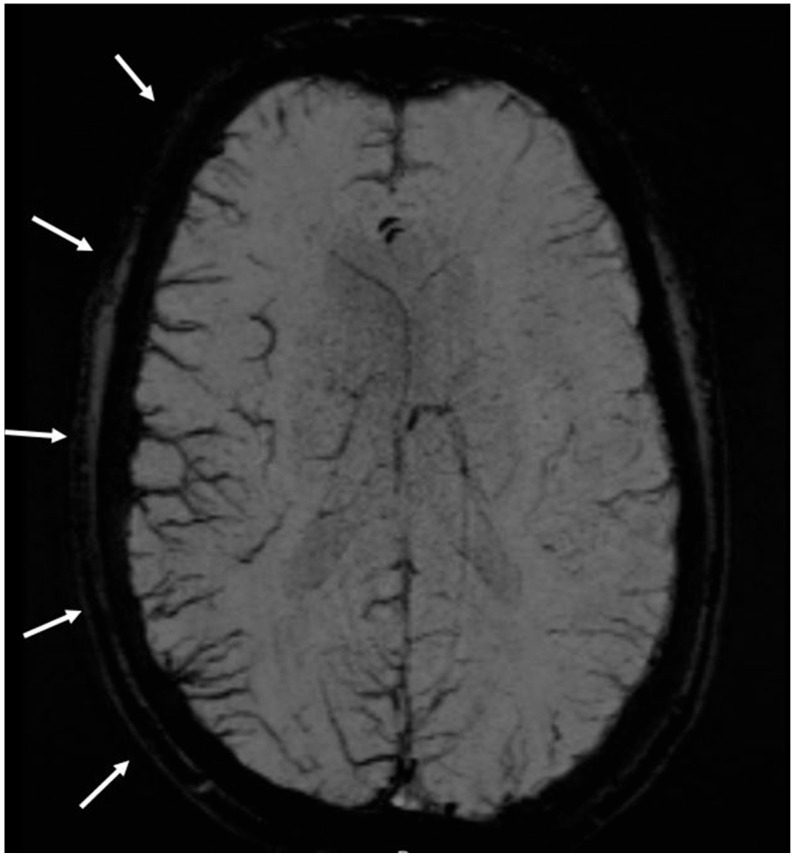
MRI susceptibility-weighted imaging (SWI), MinIP (minimum intensity projection) of a 32-year-old patient presenting to the emergency department with sensory disturbances in the left extremities and recent-onset headache. The image reveals pseudodilated cerebral veins over the right hemisphere, crossing vascular territories. Discharge diagnosis was migraine with aura.

**Table 1 neurolint-17-00104-t001:** Demographics, co-morbidities and vascular risk factors in the migraine with aura (MA), migraine without aura (MO) and controls (C).

Co-Morbidities and Vascular Risk Factors	Migraine with Aura	Migraine Without Aura	Controls	*p* Value (Bonferroni)	Post-Hoc Test		
					*p* Value MA vs. MO ^†^	*p* Value MA vs. C ^†^	*p* Value MO vs. C ^†^
age	39+/−15	36+/−11	61+/−13	< 0.001	0.1	<0.001	<0.001
female sex	68	86	55	< 0.001	0.09	0.059	<0.001
arterial hypertension	15	6	33	< 0.001	0.03	0.003	<0.001
obesity *	1	3	3	1	0.62	0.62	1
atrial fibrillation	2	0	5	0.18	0.49	0.24	0.059
diabetes mellitus	6	4	8	1	0.52	0.57	0.23
cigarette smoking	17	9	14	0.72	0.93	0.55	0.39
family history of stroke/myocardial infarction	0	1	2	1	1	0.49	1
chronic alcohol abuse ^††^	4	1	4	1	0.36	1	0.36
Illicit drugs	2	1	2	1	1	1	1
depression	2	11	8	0.12	0.01	0.052	0.46
chronic kidney failure	0	1	3	0.51	1	0.24	0.62
sleep apnea	2	0	4	1	0.49	0.4	0.12
dyslipidemia	9	3	20	<0.001	0.07	0.02	<0.001

* defined as body mass index of > 25 kg/m^2^, ^†^ Mann Whitney, Chi2 or Fisher’s exact test, ^††^ defined as >30 drinks monthly. MA = migraine with aura, MO = migraine without aura, C = controls.

**Table 2 neurolint-17-00104-t002:** Characteristics of the controls and patients with migraine without aura with SWI changes.

No	Sex	Age	Diagnosis at Discharge	Co-Morbidities	Vascular Risk Factors
1	male	49	benign positional vertigo	history of migraine without aura, fungal sinusitis ethmoidalis	arterial hypertension
2	female	71	transient global amnesia	restless-legs syndrome	none
3	male	59	transient global amnesia	none	arterial hypertension
4	female	56	benign positional vertigo	hypothyreosis, osteoarthritis	arterial hypertension, smoking, dyslipidemia
5	female	35	idiopathic facial nerve palsy	none	arterial hypertension
6	female	74	transient global amnesia	IgG (1&3) antibody deficiency syndrome	none
7	male	65	transient global amnesia	none	none
8	female	42	migraine without aura	none	depression
9	male	44	migraine without aura	none	none
10	male	52	migraine without aura	none	arterial hypertension

**Table 3 neurolint-17-00104-t003:** Symptoms in patients with migraine with aura stratified for the presence of SWI changes.

Symptoms	SWI Changes Present (N = 26)	SWI Changes Absent (N = 74)	*p* Value
Visual symptoms *n* (%)	16 (62)	49 (66)	0.66
Sensory symptoms *n* (%)	16 (62)	37 (50)	0.31
Motor symptoms *n* (%)	4 (15)	14 (19)	0.68
Speech disturbance *n* (%)	17 (65)	24 (32)	0.003
Coordination problems *n* (%)	3 (12)	1 (1)	0.053
Vertigo *n* (%)	3 (12)	11 (15)	0.67
Confusion *n* (%)	0	1 (1)	1

## Data Availability

Upon reasonable request to the corresponding author, data used for this research will be made available.

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
