# Peer review of "Susceptibility Weighted Imaging in Migraine with and Without Aura: A Case–Control Study"

_2035-8377, 2025, doi:10.3390/neurolint17070104_

Round 1
Reviewer 1 Report
Comments and Suggestions for Authors
Thank you for allowing me to review this manuscript.
--- Major Revision Required ---
The paper by Scutelnic et al underlines the presence of asymmetry of cortical veins in SWI at MRI in a cohort of patients affected by migraine with aura. The topic is interesting and quite novel, and fills some gaps in the literature, highlighting the role of neuroimaging in migraine patients as a marker of cortical spread depression. There are some methodological limitations, mainly due to the timing of the performance of MRI and the retrospective nature of the study, that are correctly addressed by the authors in the discussion of the paper. The number of references and the cited references are correct.
I have some suggestions:
- Ethics: Please specify which ethical committee approved the study, and specify the approval number (e.g. name of the ethical committee, number of the protocol)
- Ethics: "local regulations" is not specific. Authors must specify the ethical committee for the study authorisation, and the guidelines adopted to manage the patients when talking about clinical management.
- Statistical and clinical methods description must be improved and expanded in the "materials and methods section". In particular, please improve the following:
A. Specify the statistical power obtained by the sample size obtained by adding a power analysis.
B. Expand the statistical methods description, specifying better why some corrections - as Bonferroni - were adopted to improve the statistical tests results
- Recently, the same author published a similar paper in another journal (https://doi.org/10.3390/ctn9010003): similarities and differences between the two studies must be correctly addressed in the text, underlining the differences between cohorts and results.
Comments on the Quality of English LanguageEnglish language quality is acceptable, but must be improved in the revised version. A naive-speaking English corrector should help improving the final form of the paper
Author Response
Neurology International
Manuscript ID neurolint-3678736
Manuscript Title: Susceptibility weighted imaging in migraine with and without aura – a case-control study
Dear Editors, dear Reviewers,
We would like to thank you for the opportunity to revise our manuscript. We revised the manuscript accordingly. Below you will find a point-by-point answer to the editors and reviewers comments.
Reviewer 1
The paper by Scutelnic et al underlines the presence of asymmetry of cortical veins in SWI at MRI in a cohort of patients affected by migraine with aura. The topic is interesting and quite novel, and fills some gaps in the literature, highlighting the role of neuroimaging in migraine patients as a marker of cortical spread depression. There are some methodological limitations, mainly due to the timing of the performance of MRI and the retrospective nature of the study, that are correctly addressed by the authors in the discussion of the paper. The number of references and the cited references are correct.
Response: We thank the reviewer for favorably evaluating our manuscript.
Ethics: Please specify which ethical committee approved the study, and specify the approval number (e.g. name of the ethical committee, number of the protocol).
Response: We thank the reviewer for pointing this out. We wrote in the methods part of the original version of the manuscript: “This is a retrospective case-control study that was approved by the local ethics committee (Bern cantonal ethics committee, 2020-00115).”
Ethics: "local regulations" is not specific. Authors must specify the ethical committee for the study authorisation, and the guidelines adopted to manage the patients when talking about clinical management.
Response: We thank the reviewer for raising these points. We now specify the regulations that allowed us to analyze patient-related data: “The written general consent allows patient-related data from the routine clinical care to be used in anonymized form for research purposes.”
Statistical and clinical methods description must be improved and expanded in the "materials and methods section". In particular, please improve the following:
A. Specify the statistical power obtained by the sample size obtained by adding a power analysis.
Response: We thank the reviewer for this comment. Our research is an exploratory analysis, and given the lack of previous data on the prevalence of SWI changes in migraine without aura, no prior power analyses were performed. We added these aspects to the manuscript: “Due to the exploratory nature of this analysis and lack of data on frequency of SWI changes in migraine without aura, no prior power calculation was performed.”
- Expand the statistical methods description, specifying better why some corrections - as Bonferroni - were adopted to improve the statistical tests results.
Response: Point well taken. We added to the methods: “For multiple comparisons, Bonferroni correction has been performed, to correct for erroneous inferences and for stricter significance thresholds.”
Recently, the same author published a similar paper in another journal (https://doi.org/10.3390/ctn9010003): similarities and differences between the two studies must be correctly addressed in the text, underlining the differences between cohorts and results.
Response: The paper mentioned by the reviewer differs in multiple ways from the current research:
- In the published paper only migraine with aura were included (n=18) vs three groups in the current paper (n=100 per group).
- In the published paper we analyzed the distribution of the SWI changes in distinct brain regions in patients with SWI changes vs frequency of SWI changes in three different groups.
- The research question were fundamentally different: in the published paper we test the hypothesis that CSD occurs in the posterior regions of the brain, whereas in the current paper the explore the question whether the CSD occurs in the migraine without aura, two decisively distinct research questions.
If the editors and reviewer agree, given the distinct hypotheses, research questions, methods and included populations, and therefore lack of overlap, we would not include these aspects in the current paper.
English language quality is acceptable, but must be improved in the revised version. A naive-speaking English corrector should help improving the final form of the paper
Response: We thank the reviewer for this criticism. We now improved the quality of the written English in this paper.
Reviewer 2 Report
Comments and Suggestions for Authors
The article under review analyzes the interhemispheric asymmetry of cerebral veins, as measured by SWI imaging, in large cohorts of patients with migraine with aura, compared to patients with migraine without aura and to controls. The subject is interesting, as SWI-measured pial/cortical veins dilation is considered an indirect marker of cortical spreading depression in patients with migraine; the interrelations between CSD, clinical aura, vasodilation and headache development are still unclear, with controversial data from studies as detailed in the manuscript. I believe the paper is a contribution to positive knowledge on the subject, but I would suggest some further clarification and discussion on certain aspects.
- One major difference between groups is the timing of the study in migraine patients without aura. By inclusion criteria, these patients were studied during the migraine headache; at the same time the interquartile range is 26 to 216 hours after symptom onset. This would mean that the bigger part of them had migraine status continuing over one to ten or more days? This seems to me unlikely, unless those were patients with chronic migraine and daily headache for weeks on end?
- The authors suggest that timing of the study is not significant for detection of venous asymmetry, because the latter (the asymmetry) persisted even after cessation of symptoms in migraineurs without aura and in controls with focal neurologic deficit. This logic seems applicable in limited time intervals only.
- The second question for me is – what treatment was applied and whether treatment could somehow influence the results of this study. Vasoconstriction is the expected effect of abortive medications, although it is believed that venoconstriction at therapeutic doses is less pronounced than extracerebral arterial constriction; it might be useful to report on the treatment of the patients and to comment on its possible effect.
- Vessel asymmetry is easily appreciated, when one observes images like the illustration presented. However, when involvement of 2 veins only is suggested sufficient, is there any numeric ratio between affected and unaffected vessels? How dilated should the veins be to consider the asymmetry abnormal?
- Some details seem unexpected. E.g. – out of 100 patients per group just 1 to 3 were overweight, while for Switzerland the reported percent is above 40% in the general population. Again, only 0 to 1% had any family history of cerebrovascular and cardiovascular disease. Were these indeed unusually slim and healthy cohorts.
In all, I believe the study is a valuable contribution, and I expect the authors to clarify the question regarding study done during the headache but several days to weeks after its onset.
Author Response
Neurology International
Manuscript ID neurolint-3678736
Manuscript Title: Susceptibility weighted imaging in migraine with and without aura – a case-control study
Dear Editors, dear Reviewers,
We would like to thank you for the opportunity to revise our manuscript. We revised the manuscript accordingly. Below you will find a point-by-point answer to the editors and reviewers comments.
Reviewer 2
The article under review analyzes the interhemispheric asymmetry of cerebral veins, as measured by SWI imaging, in large cohorts of patients with migraine with aura, compared to patients with migraine without aura and to controls. The subject is interesting, as SWI-measured pial/cortical veins dilation is considered an indirect marker of cortical spreading depression in patients with migraine; the interrelations between CSD, clinical aura, vasodilation and headache development are still unclear, with controversial data from studies as detailed in the manuscript. I believe the paper is a contribution to positive knowledge on the subject, but I would suggest some further clarification and discussion on certain aspects.
Response: We thank the reviewer for the favorable evaluation of our manuscript.
One major difference between groups is the timing of the study in migraine patients without aura. By inclusion criteria, these patients were studied during the migraine headache; at the same time the interquartile range is 26 to 216 hours after symptom onset. This would mean that the bigger part of them had migraine status continuing over one to ten or more days? This seems to me unlikely, unless those were patients with chronic migraine and daily headache for weeks on end?
Response: Thank you for raising this important point. Indeed, many of the included patients had status migrainosus, with headaches which did not remit after multiple treatments. Usually the presence of headache of longer duration in patients with known migraine does not require emergency imaging. However, in our population due to the longer duration of headache, there was suspicion of secondary causes such as cerebral venous thrombosis. In this sense, there might have been a selection bias of patients with headache attacks of longer duration, hence explaining the longer time interval between headache onset and MRI. We now added these aspects to the manuscript. “There might have been a selection bias of patients with headache of longer duration, given the clinical suspicion of secondary headache which requires MRI examination.”
The authors suggest that timing of the study is not significant for detection of venous asymmetry, because the latter (the asymmetry) persisted even after cessation of symptoms in migraineurs without aura and in controls with focal neurologic deficit. This logic seems applicable in limited time intervals only.
Response: We agree with the reviewer, that this observation might be applicable in limited time intervals only. However, given the cross-sectional design and availability of one MRI study only, we could not assess the time point when the SWI changes become reversible. Further longitudinal studies with repeat MRI examination at different time points would be necessary to address this. We now added this aspect to the manuscript: “This finding, however, needs confirmation - ideally through longitudinal studies involving repeat MRIs at different time points - since previous reports found an association between venous changes assessed by MRI and time from symptom onset in migraine with aura.”
The second question for me is – what treatment was applied and whether treatment could somehow influence the results of this study. Vasoconstriction is the expected effect of abortive medications, although it is believed that venoconstriction at therapeutic doses is less pronounced than extracerebral arterial constriction; it might be useful to report on the treatment of the patients and to comment on its possible effect.
Response: Point well taken. Due to heterogeneity of daily and migraine abortive medication and lack of precise documentation of the timepoint of medication intake, we did not assess the possible influence of treatment on our findings. We now added this in the limitation part of the manuscript.
Vessel asymmetry is easily appreciated, when one observes images like the illustration presented. However, when involvement of 2 veins only is suggested sufficient, is there any numeric ratio between affected and unaffected vessels? How dilated should the veins be to consider the asymmetry abnormal?
Response: We thank the reviewer for raising this point. SWI images were first visually screened for unilateral prominence of cortical veins by the rating neuroradiologists. When ≤2 veins appeared prominent as rated by the expert neuroradiologist, we applied the approach described in the study by Slavova et al. (Neurology 2020). The maximal diameter of the most prominent vein was measured on the source SWI image and divided by the diameter of its contralateral homologue as originally applied for the definition of index vein. A ratio ≥ 2.0 was considered significant. When >2 veins appeared prominent after visual analysis, the venous asymmetry was graded qualitatively (mild/moderate/marked) by the two blinded neuroradiologists (VP and TK) in the predefined regions as described in previous studies. Discrepancies were resolved in consensus.
We have amended the Methods section (page 6) accordingly:
“When ≤2 veins appeared prominent as rated by the expert neuroradiologist, we applied the approach described in the study by Slavova et al. (Neurology 2020). The maximal diameter of the most prominent vein was measured on the source SWI image and divided by the diameter of its contralateral homologue. A ratio ≥ 2.0 was considered significant. When >2 veins appeared prominent after visual analysis, the venous asymmetry was graded qualitatively (mild/moderate/marked) by the two blinded neuroradiologists (VP and TK) in the predefined regions as described in previous studies. Discrepancies were resolved in consensus.”
Some details seem unexpected. E.g. – out of 100 patients per group just 1 to 3 were overweight, while for Switzerland the reported percent is above 40% in the general population. Again, only 0 to 1% had any family history of cerebrovascular and cardiovascular disease. Were these indeed unusually slim and healthy cohorts.
Response: We agree that some baseline characteristics do not correspond to what would have been expected in an unselected general population. This is possible due to the retrospective design and possibly lack of systematic assessment in the emergency setting. We now expand on limitations: “Another limitation of our study is its retrospective design and lack of systematic assessment of clinical data (e.g. of baseline characteristics such as obesity).”
In all, I believe the study is a valuable contribution, and I expect the authors to clarify the question regarding study done during the headache but several days to weeks after its onset.
Response: We thank the reviewer for the favorable evaluation. The question regarding the MR study done several days up to 10 days after headache onset has been addressed above.